# Metabolic Modeling of *Hermetia illucens* Larvae Resource Allocation for High-Value Fatty Acid Production

**DOI:** 10.3390/metabo13060724

**Published:** 2023-06-03

**Authors:** Kristina Grausa, Shahida A. Siddiqui, Norbert Lameyer, Karin Wiesotzki, Sergiy Smetana, Agris Pentjuss

**Affiliations:** 1Department of Computer Systems, Latvia University of Life Sciences and Technologies, LV-3001 Jelgava, Latvia; kristina.grausa@gmail.com; 2Institute of Microbiology and Biotechnology, University of Latvia, LV-1050 Riga, Latvia; 3Campus Straubing for Biotechnology and Sustainability, Technical University of Munich, Essigberg 3, D-94315 Straubing, Germany; s.siddiqui@dil-ev.de; 4German Institute of Food Technologies (DIL e.V.), 49610 Quakenbrück, Germany; n.lameyer@dil-ev.de (N.L.);

**Keywords:** metabolic modeling, insects, fatty acids, *Hermetia illucens*, resource allocation

## Abstract

All plant and animal kingdom organisms use highly connected biochemical networks to facilitate sustaining, proliferation, and growth functions. While the biochemical network details are well known, the understanding of the intense regulation principles is still limited. We chose to investigate the *Hermetia illucens* fly at the larval stage because this stage is a crucial period for the successful accumulation and allocation of resources for the subsequent organism’s developmental stages. We combined iterative wet lab experiments and innovative metabolic modeling design approaches to simulate and explain the *H. illucens* larval stage resource allocation processes and biotechnology potential. We performed time-based growth and high-value chemical compound accumulation wet lab chemical analysis experiments on larvae and the Gainesville diet composition. We built and validated the first *H. illucens* medium-size, stoichiometric metabolic model to predict the effects of diet-based alterations on fatty acid allocation potential. Using optimization methods such as flux balance and flux variability analysis on the novel insect metabolic model, we predicted that doubled essential amino acid consumption increased the growth rate by 32%, but pure glucose consumption had no positive impact on growth. In the case of doubled pure valine consumption, the model predicted a 2% higher growth rate. In this study, we describe a new framework for researching the impact of dietary alterations on the metabolism of multi-cellular organisms at different developmental stages for improved, sustainable, and directed high-value chemicals.

## 1. Introduction

In recent years, a consensus has been reached regarding the fact that the resource allocation strategies and mechanisms present in different growth and dietary conditions are key aspects in the determination of an organism’s phenotypic specificities [1,2], including metabolism and even biomass composition [3,4,5].

The insect-rearing sector is rapidly growing in Europe and is becoming one of the sustainable choices for the production of feed, food, and other substances. To ensure the efficiency and sustainability of insect production, the production should rely on waste and under-utilized side streams from agri-food systems [6,7]. At the same time, there is currently no specific chemically defined diet which can be used as a reference and a basis for the cultivation of mass-produced insects. The current studies rely on the Gainesville diet, which has an approximate chemical composition and can have variations in the specific chemicals in the range of 20–30%. Multiple studies using this diet as a reference obtain different, sometimes opposite results [8,9].

*Hermetia illucens* (black soldier fly, BSF) has vast potential in the industry due to its ability to rapidly recycle organic waste and decompose the material into high-value lipids and proteins [10]. During the larval stage, the BSF rapidly accumulates nutrients from the environment and increases its body weight according to a sigmoidal growth function [11]. The BSF has gained significant interest in the fields of food and feed; biofuels; pharmaceutical, lubricant, and fertilizer sciences; and industrial applications [12]. The unbalanced fatty acid content of BSF larvae, however, is one of the main challenges faced when attempting to provide consistency and predictability in their industrial applications [13], [14]. This warrants further investigation and metabolic modeling with regard to the dietary impact on larvae body fat composition and growth rate in order to estimate their potential use for the upcycling of different agricultural waste products and residues. Recent studies determining the impact of specific nutrients on the development of *H. illucens* indicated that they indeed had an impact, and that the diets of insects would require high amounts of proteins [15] and lipids [16]. However, the studies do not provide a detailed model for the optimal chemically defined diet. Such a diet would allow for improvements in the experimental and theoretical studies, the optimization of insect production, and the identification of suitable waste and side-stream mixes.

Metabolic models successfully describe genotype–phenotype relationships for prokaryotic, eukaryotic, unicellular, and multi-tissue organisms, e.g., *Escherichia coli* [17] and *Saccharomyces cerevisiae* [18]. Metabolic modeling involves the analysis of an organism’s phenotypic responses [19]. Genome-scale metabolic modeling (GSM) has led to significant breakthroughs in the fields of health and systems medicine [20]; in new nutrient uptake innovations [20]; in different novel software developments [21,22]; in new biotechnology applications [23,24,25]; and in other improvements in the life science fields [26]. Genome-scale metabolic models were also used to find new bio-based technologies to decrease waste streams and pollution [27], to fine-tune genetic engineering design sets, and to explain an organism’s metabolic processes in different environments [27]. These models can fill the knowledge gaps relating to phenotype–genotype relationships and biochemical network connectivity by combining in vivo experimental and publicly available omics datasets.

Several insect models already exist, such as central nervous system energy metabolism [28], the cost of metabolic interaction between insects and bacteria [29], evolutionary trajectories for bacterial and insect symbionts [30], and many more. To date, however, there has been no published *H. illucens* in silico mathematical metabolic model which could describe the different resource allocation processes based on gene-protein-reaction association, genetics, feed diet, and environmental changes.

Humans and animals are heterotrophs, and they need to consume other organisms to accumulate nutrients which cannot be synthesized within the body but are crucial for metabolic processes and health. Insect larvae show great potential in the battle against the growing shortages of essential nutrients, especially fatty acids for food and feed [31].

We developed the first high-quality, manually curated, medium-scale BSF larvae metabolic model, Hermetia_01, and applied it to the modeling of high-value fatty acid production and accumulation. The model incorporates experimentally measured dietary nutrient data (amino acids and monosaccharides) on the uptake rates and the accumulated biomass macromolecular composition, including the measured fatty acid profile. The lauric acid (C12:0) content in the BSF larvae biomass was 29% of the total fatty acid profile, which makes it the most abundantly found fatty acid in *H. illucens* larvae, according to our chemical analysis. The oleic acid (C18:1) content in the BSF larvae biomass was 21% of the total fatty acid profile, and the linoleic acid (C18:2) content was 20%. The model was adjusted to simulate the accumulation, storage, and resource allocation of the most abundant *H. illucens* fatty acids: lauric acid (C12:0), myristic acid (C14:0), palmitic acid (C16:0), palmitoleic acid (C16:1), stearic acid (C18:0), oleic acid (C18:1), and linoleic acid (C18:2).

The steady-state assumption is frequently used in metabolic modeling and serves as a means of validating the interconnectivity of a biochemical network and the feasibility of its parameters. This assumption predicts that internal metabolite production and consumption must be balanced [32]. Many metabolic modeling analysis methods incorporate the steady-state assumption in their formulation; one such method is flux balance analysis (FBA) [33]. It is a constraint-based linear programming method which is dependent on the constraints enforced by the metabolic network stoichiometry and is the most frequently used method for metabolic network analysis.

## 2. Materials and Methods

### 2.1. Larvae-Rearing Experiments

Newly hatched 9-day-old larvae from eggs of the black soldier fly, *H. illucens* (Diptera: *Stratiomyidae*), were used for the experiments and reared on chicken feed (GS Gänse-Mastfutter 1, metabolizable energy content 12, 1 MJ ME, 70% moisture) for around 7 days. The larvae were at the V instar larval stage.

They were then transferred to a substrate containing 10% glucose, 20% egg white protein, and 70% water. The experiment was conducted in a plastic box (Transoplast, 60 cm × 40 cm × 12.5 cm) and kept at 27 °C and 60% humidity. Four hundred and fifty grams of larvae were placed in each box. The experiment was stopped after nine days of feeding. The substrate was weighed when applied, as was the total remaining material at the end of the experiment.

### 2.2. Larvae Wet Weight Measurements

For the single larva weight, the combined weight of 30 larvae was recorded and divided by 30 to calculate the single larva weight. The larvae were separated from their feed. At the end of the experiment, the larvae were collected, then placed in a sieve and thoroughly washed with water with extreme care, dried on a piece of paper, and finally weighed.

### 2.3. Experimental Chemical Analytics and Analysis Methods

#### 2.3.1. GC-MS Measurements

All the studies were conducted with the help of the Hewlett-Packard 5890 (GC)/5970 mass selective detector (MSD) in the electron impact (EI) mode (70 eV), using a configuration setup with a fused silica capillary column (HP-5ms, 30 m × 25 mm ID, 0.25 µm film thickness; Agilent J&W Scientific, Folsom, CA, USA). The 2 µL sample was inserted using the splitless mode. The cover was kept at a temperature of 70 °C for 3 min. Afterwards, it was increased to 280 °C at a rate of 25 °C/min and then kept at this temperature for 5 min. Helium was used as the transporting gas and was fed in at a constant rate of 1 mL/min. The injector temperature was kept constant at 240 °C, and the terminal temperature was kept at 290 °C. The mass spectrum of the MCF-derivatized amino acids and the inherent criteria were gathered in either the full-scan (50–350 m/z) or the selected ion monitoring (SIM) acquisition modes. Amino acid measurements were conducted by AGROLAB LUFA.

#### 2.3.2. Fatty Acid Profile

In a modification, after freeze-drying, the samples were heated directly with a methanolic sulfuric acid solution. The resulting fatty acid methyl esters were then extracted using *n*-hexane and analyzed by capillary gas chromatography, according to the standard procedure. The determination was done according to the ISO 12966-2 standard Section 5.5.

#### 2.3.3. Sugars (Glucose, Fructose, Sucrose, and Maltose) and Glycerol

The sugars were measured using HPLC and a refractometric detector in accordance with AOAC 982.14. The necessary sample preparation for this was carried out in accordance with L05.00-10, Section 7.2.1 (Federal Office for Consumer Protection and Food Safety, Germany, the official collection of examination procedures) (5 g sample, Carrez clarification).

#### 2.3.4. Starch

The method comprises two determinations. In the first, the sample was treated with dilute hydrochloric acid. After clarification and filtration, the optical rotation of the solution was measured by polarimetry. In the second, the sample was extracted with 40% ethanol. After acidifying the filtrate with hydrochloric acid, then clarifying and filtering, the optical rotation was measured as in the first determination. The difference between the two measurements, multiplied by a known factor, gives the starch content of the sample. The determination was in accordance with Appendix III L of Regulation (EC) No. 152/2009, which establishes the methods of sampling and analysis for the official inspection of feed.

#### 2.3.5. Fat

With the addition of an internal standard, fat from the samples was extracted and, at the same time, the alkaline was digested. After extraction, the potassium salts were converted into free fatty acids, and the proportion of fatty acids was determined by gas chromatography. The fat content, calculated as triglyceride content in a 100 g sample, was calculated using response and conversion factors and a calibration standard for fat. The determination was in accordance with AOAC Method PVM4: 1997 (Caviezel^®^).

### 2.4. Metabolic Network Reconstruction of Hermetia Illucens Larvae

Insects are multi-tissue, multi-organ animals, and the metabolic modeling of such organisms can be challenging. Extensively detailed genome-scale metabolic networks, which describe whole-body systems (e.g., human and *A. thaliana*), can offer new and profound insights into organism metabolism. However, the development of such models requires decades-long research, extensive literature, and a knowledge base of the target organism, its organ systems and genetics, and other details and specificities.

We generated a draft reconstruction based on a genome-scale model development protocol using *H. illucens* genome annotation data (available on (12 October 2022): https://www.ncbi.nlm.nih.gov/assembly/GCF_905115235.1). A draft model was generated with the RAVEN [34] metabolic modeling tool. Next, we applied manual reaction curation using biochemical databases such as MetaCyc [35], BioCyc [35], KEGG [36], PubChem [37], and ChEBI [38]; in addition, we used previously published and curated *Drosophila melanogaster* larval central metabolic model data, focusing on the pathways covering our quantitative experimental measurements and fatty acids. Our metabolic network covered a total of 326 metabolites, 407 reactions, 471 genes, and 3 compartments, and the model scale included extended central metabolism pathways which were analogues of the functionality of the *E. coli* central metabolism [39]. The interactive metabolic model visualization was drawn by using the IMFLer online modeling tool [40] (Appendix A) (Figure 1).

To recharge ferricytochrome b5, which is used for oleic and linoleic acid synthesis, we included an additional reaction called the cytochrome-b5 reductase reaction (BioCyc ID: CYTOCHROME-B5-REDUCTASE-RXN). Furthermore, we extended our metabolic model by incorporating a diacylglycerol and triacylglycerol biosynthesis pathway (TRIGLSYN-PWY) according to the redGEM approach [41] and by lumping together several reactions in a single one, in order to model the fat storage of TAG (12:0/14:0/16:0) and the TAG (16:1 or 18:0/18:1/18:2) within the BSF larvae biomass.

The metabolic network was reconstructed using the MATLAB environment, and the optimization, analysis, and validation tasks were performed using Cobra Toolbox 3.0 software [42]. We also integrated our experimental measurements on the transport reactions, which represented the substrate in which the larvae were grown and the given diet during the wet lab experiments.

### 2.5. The Biomass Metabolic Reaction

We used quantitative BSF larvae chemical analysis experiments on the amino acids, fat (triglyceride), starch (glycogen), sugars (mono- and disaccharides), and glycerol to calculate their respective ratios per one dry weight (DW) larva. As the cholesterol concentration was not captured during the wet lab experiments but is a major constituent of animal cell membranes, we approximated the ratio of cholesterol relative to the other biomass metabolites using a previously defined and published *D. melanogaster* larvae biomass function. Chemical analysis experiments on the water, carbohydrates, and ash were omitted from the biomass composition, and the rest of the biomass macromolecule ratios were normalized to 100% of the biomass. The biomass metabolite ratios (Section 3.2) were then converted to mmol per one BSF larva DW and used in the biomass reaction definition.

We selected TAG (12:0/14:0/16:0) and TAG (16:1/18:1/18:2) in our biomass reaction formulation, with a ratio of 49:51 of the total biomass fat, respectively. These triglycerides were chosen because they comprise the fatty acids that are the most abundant in BSF larvae according to our chemical analysis data of fatty acid composition; together, they account for 90% of the total fatty acids.

### 2.6. Lipid Metabolism

We focused on lipid metabolism and specific fatty acid (C12:0, C14:0, C16:0, C16:1, C18:0, C18:1, C18:2) synthesis while developing a metabolic model of the BSF larvae. These fatty acids were of special interest in the modeling of *H. illucens* larvae metabolism due to their high content in the organism, as captured by fatty acid profiling measurements. They are also of interest because of their biotechnological significance and the prospects for the optimizing, analyzing, and predicting of their yields when using bioconversion for recycling and the reduction in industrial and agricultural waste [43].

The lipid metabolism reaction complexes included in our metabolic model cover several pathways:Fatty acid biosynthesis initiation (type I), PWY-5966;Palmitate biosynthesis I (type I fatty acid synthase), PWY-5994;Tetradecanoate biosynthesis (mitochondria), PWY66-430;Stearate biosynthesis I (animals), PWY-5972;Oleate biosynthesis II (animals and fungi), PWY-5996;Linoleate biosynthesis II (animals), PWY-6001.

The consecutive biochemical reactions listed in the pathways above were collated into lumped reactions (substrates and product metabolites), with one reaction for each respective Acyl-CoA synthesis coupled with a hydrolase reaction to synthesize the respective fatty acid. Oleoyl-CoA and linoleoyl-CoA synthesis required a cytochrome-b5 reductase reaction (BioCyc ID: CYTOCHROME-B5-REDUCTASE-RXN), which was also added to the model. Consequently, we extended our metabolic model by incorporating diacylglycerol and triacylglycerol biosynthesis pathway (TRIGLSYN-PWY) reactions (EC: 2.3.1.15, 2.3.1.51, 2.3.1.23, 3.1.3.4, 2.3.1.20) for TAG (12:0/14:0/16:0) and TAG (16:1/18:1/18:2) to model the storage of the different fatty acids in the BSF larvae biomass.

### 2.7. Approximation of Food Intake and Transport Reaction Constraints

The *H. illucens* larvae received 8 kg of food in 9 days; the diet was composed of 5284 g of water (Table 1) and a total of 846 g glucose and 1870 g egg white protein with an amino acid composition, as listed in Table 2. No glucose was left in the substrate at the end of the 9 days, but there was protein and water residue, which was measured during the wet lab experiments. We assumed a linear amino acid and glucose consumption during the 9 days of larval growth.

The dietary amino acid profile was converted to transport reaction rates per 1 larva a day (mmol/1 larva/day), while considering the amino acid quantity left in the residue, as shown in our chemical analysis experiments on the BSF larva amino acid composition (Table 2) (more detailed information is found in Appendix A). We determined that the larvae count after the rearing experiment was 11.25 × 10^6^ and that the larvae total weight was 225 g, which allowed us to calculate that the approximate larva weight on the last day of experiment was 0.028 g.

### 2.8. Data Analysis

To observe and assess the resource allocation changes within our metabolic network based on the applied diet, we performed two sets of in silico experiments: (a) a simulation of the biomass growth rate changes for the reference state based on our experimental data and under the conditions of increased glucose or essential amino acids (EAA) and (b) a simulation of the different fatty acid (C12:0, C14:0, C16:0, C16:1, C18:0, C:18:1, C18:2) synthesis rate changes with increased glucose consumption and an estimation of the dietary carbon conversion efficiency (*CCE*, Equation (1)). During the first set of in silico analyses in the MATLAB environment using the Cobra Toolbox FBA method, a steady state for our experimental data constraints was confirmed, and the biomass-specific growth rate was optimized under conditions of doubled glucose or EAA uptake. The objective function during FBA maximization was consequently set to the biomass growth reaction.

The second set of experiments involved the use of FVA, preserving 90% of the maximal DW biomass-specific growth rate per day, as described by [44] for fatty acid (C12:0, C14:0, C16:0, C16:1, C18:0, C18:1, C18:2) synthesis rate modeling under doubled glucose uptake conditions. Afterwards, the dietary carbon conversion efficiency was estimated using Formula 1, and both states (reference and doubled glucose uptake) were compared.
(1)$$CCE=\frac{\sum_{i=1}^{n}\left(C_{i}*uptake\_rate\right)}{C_{f}*drain\_rate}*100$$

Equation (1). Carbon conversion efficiency (*CCE*) in %, estimated using summary diet uptake reaction rates (*uptake_rate*) times carbon atoms per molecule (calculated by FBA in mmol/larva/day), divided by the maximal theoretical fatty acid synthesis rate (*drain_rate*) times carbon atoms per molecule (calculated by FVA in mmol/larva/day) multiplied by 100, where

*n*—exchange reaction count;

*Ci*—carbon atom count in a dietary nutrient (amino acids, glucose, and cholesterol);

*Cf*—carbon atom count in the fatty acid molecule.

## 3. Results

### 3.1. The First Medium-Scale Metabolic Model for Hermetia Illucens

Considering that no metabolic model of *H. illucens* larvae has previously been published despite the growing interest in its ability to rapidly recycle organic waste and decompose the material into high-value lipids and proteins [14,45,46], we generated and manually curated Hermetia_01 (available in JSON, MAT, XLSX, and SBML file formats in Appendix A), which describes the metabolism of *H*. *illucens* at the larval development stage. Our model covers a total of 326 metabolites, 407 reactions, 471 genes, and 3 compartments; it provides a way to study the biochemical pathways involved in fatty acid and amino acid synthesis by BSF larvae in order to assess their potential as a sustainable source of lipids and proteins, not only for food and feed but also for pharmaceutical products [47,48,49]. The unbalanced fatty acid content of BSF larvae is one of the main challenges facing their use in food and feed [14]. For this purpose, we focused on lipid metabolism and specific fatty acid (C12:0, C14:0, C16:0, C18:0, C18:1, C18:2) synthesis and accumulation in the biomass during the development of the metabolic model. The six previously mentioned fatty acids were selected because, taken together, they represented ~90% of the total fatty acid composition of BSF larvae, based on our chemical analysis. Hermetia_01 was tailored for modeling, predicting, and optimizing fatty acid and amino acid metabolism processes and resource allocations when using different diets and substrates.

### 3.2. Biomass Composition and Reaction

The given diet, substrate composition, and environmental factors affect the growth and development of any organism and result in varying biomass compositions which are specific to that environment. Some fatty acids, mostly saturated (C12:0, C14:0, C16:1), were shown to be synthesized by BSF larvae and not bioaccumulated [48,50].

Using our chemical analysis experiments on the amino acid, fatty acid, and macromolecular composition of *H. illucens* larvae, we formulated a basic level biomass objective function [51], which describes the composition of the cell and the energetic requirements necessary to generate the biomass content from metabolic precursors and to sustain cell proliferation. Our biomass objective function (Figure 2) consisted of amino acids, fat (triglyceride), starch (glycogen), sugars (mono- and disaccharides), and glycerol, with respective ratios in mmol per one dry weight (DW) larva, calculated based on our experimental measurements (Appendix A). The cholesterol was not captured during chemical analysis but is a major constituent of animal cell membranes; therefore, it was approximated relative to other biomass macromolecules using a previously published *D. melanogaster* metabolic network [52]. We normalized our DW biomass macromolecule ratios to 100%.

Triglycerides (TAG) are esters derived from glycerol and three fatty acid chains of varying lengths and compositions. These esters are the main components of animal fat and are stored in cellular organelles called lipid droplets or adiposomes [53], inside specialized storage organs such adipose tissue in mammals or the fat body in insects [54]. In the case of holometabolous insects, such as the BSF, fat storage is essential during the larval development stage as triglycerides serve as an energy storage and resource [55]. For modeling purposes, we first estimated the most abundant fatty acids in BSF larvae based on our chemical measurements of fatty acid composition (Table 3); these were 29% of C12:0; 6% of C14:0; 10% of C16:0; 6% of C16:1; 21% of C18:1; and 20% of C18:2. Taken together, these fatty acids made up 90% of the total fatty acid composition in the BSF larvae (Appendix A) (Table 1). Our chemical measurements of the larvae fatty acid composition indicated 49% saturated fatty acids (C12:0, C14:0, C16:0, C18:0) and 48% unsaturated fatty acids (C14:1, C16:1, C18:1, C18:2), of which 28% were monounsaturated (C14:1, C16:1, C18:1) and 20% were polyunsaturated fatty acids (C18:2). Consequently, we extended our metabolic model by incorporating the diacylglycerol and triacylglycerol biosynthesis pathway (TRIGLSYN-PWY) for TAG (12:0/14:0/16:0) and TAG (16:1/18:1/18:2) synthesis, which we used in the biomass objective function formulation for BSF larvae, with a ratio of 49:51 of the total biomass fat, respectively (Figure 2).

### 3.3. Model Validation and Evaluation

We validated the feasibility of our metabolic network and the applied constraints by performing FBA on the biomass function as the optimization objective and by having the transport reactions constrained according to the given diet during wet lab experiments. The steady state was reached and this allowed us to extract the maximal theoretical biomass growth rate.

Next, we evaluated our metabolic model based on several parameters described in a previously published protocol for generating a high-quality genome-scale metabolic reconstruction [53]. We used Cobra Toolbox functions on our metabolic network in the MATLAB environment and estimated 2.45% dead-end metabolites (*detectDeadEnds*), 16.71% blocked reactions (*findBlockedReaction*), 19.66% unbalanced reactions (*checkBalance*), and 18.43% exchange reactions (*findExcRxns*) within our model, and we compared the results with those of other metabolic networks (Table 4).

We also validated our DW biomass fat (16%) and amino acid (70%) ratios of the BSF larvae; these ratios were in line with those of the previous literature [10]. The fatty acid profile measurements, which consisted of lauric acid (C12:0) 29%, myristic acid (C14:0) 6%, palmitic acid (C16:0) 10%, palmitoleic acid (C16:1) 6%, oleic acid (C18:1) 21%, and linoleic acid (C18:2) 20%, were within the exact range described in the previous literature [11].

### 3.4. Metabolic Flux Potential as Predicted by Flux Variability Analysis (Resource Allocation)

To assess the differences in the resource allocation mechanisms within the Hermetia_01 metabolic network of the BSF larvae based on the given diet, we performed several in silico experiments utilizing the flux variability analysis (FVA) [56] provided by the Cobra Toolbox software. The FVA method calculates maximal and minimal biochemical network reaction rate values for each reaction in a metabolic network while maintaining (in our case) 90% of the maximal biomass-specific growth rate per day, as described by [44].

We used the novel metabolic model, Hermetia_01, for the execution of two sets of in silico BSF larvae metabolism experiments: (1) the prediction of specific growth rate changes based on dietary alterations (under the conditions of doubled glucose or doubled essential amino acid uptake) and (2) the prediction of fatty acid (C12:0, C14:0, C16:0, C16:1, C18:0, C:18:1, C18:2) synthesis rate changes with doubled glucose uptake and the calculation of dietary carbon conversion efficiency. All the results are in DW.

In the growth rate change experiments, we defined three different design sets to obtain the theoretical substrate carbon distribution potential and to predict the impact of the substrate (glucose and amino acids) uptake change on the organism’s specific growth rate. (1) Doubling the glucose consumption rate in the Hermetia_01 medium-size metabolic model did not show any increase in the specific growth rate when compared to the unchanged dietary state based on our experimental data for the BSF larvae; these results were similar to those of the predictions of the *FlySilico* model [52] of *D. melanogaster* larvae.

Mammals are heterotrophic organisms which require the consumption of specific nutrition, such as EAA, through their diets. We prepared an optimization design set, where (2) we doubled all the EAA consumption rates (Figure 3).

The Hermetia_01 model showed the potential to increase the BSF-specific growth rate by 32% (Figure 4). The model could not simulate all amino acid degradation metabolism processes, but it doubled the valine consumption and the later degradation in the chemical building blocks, resulting in a 2% increase in the specific growth rate.

The fatty acid (C12:0, C14:0, C16:0, C16:1, C18:0, C18:1, C18:2) synthesis rates increased when the glucose consumption rate was doubled in the Hermetia_01 model (Figure 5), particularly that of C18:2, which increased from 0.029 to 0.051 mmol/larva/day, or 75.86%, compared to the reference diet.

However, the conversion efficiency of dietary carbon to high-value fatty acids dropped for the doubled glucose consumption diet in comparison to the reference diet. This indicated that the additional amount of nutrition was not used sufficiently (Figure 6) for fatty acid synthesis and was diverted to other metabolic needs when the production threshold was reached. One exception was C18:2, for which the doubled glucose diet increased the dietary carbon conversion efficiency by 3.25% compared to the reference diet.

The performed growth rate change in the in silico experiments indicated that the low growth rate of the BSF larvae was caused by amino acid deficiencies rather than the lack of energy resources (Figure 4); this is in line with the prognosis of other researchers studying the dietary impact on *H. illucens* larva metabolism and growth [57]. Therefore, to optimize BSF larvae biomass production, additional protein supplementation in the diet should be considered.

As for optimizing fatty acid (C12:0, C14:0, C16:0, C16:1, C18:0, C:18:1, C18:2) synthesis and accumulation in the BSF larvae biomass, an optimal glucose uptake threshold should be determined and tested during future wet lab experiments, while considering the dietary carbon conversion efficiency changes when glucose uptake is increased.

This knowledge can be applied to adjust the BSF larvae diet composition, to increase the biotechnological potential of different product metabolites, and to determine the applicable waste products or residues for BSF larvae to recycle.

## 4. Discussion

Even though it is currently strictly regulated, insect use in food and feed has attracted the attention of many researchers worldwide due to the rich protein and fat composition of many insects [52,58]. The potential of insect utilization for agricultural waste recycling, reuse, and supplementation in feed is particularly relevant for farmers, food, and feed manufacturers, and, potentially, pharmaceutical manufacturers. There are several important advantages of insect rearing compared to livestock farming; these include reduced resource requirements, high feed conversion efficiency, and the potential for the recycling and reuse of industrial side streams or byproducts, [58]. Apart from the high-quality protein content necessary for feed, fatty acids serve as energy sources for heterotrophic organisms in the world. In this study, we focused on those fatty acids which are found most abundantly in BFS compositions (C12:0, C14:0, C16:0, C16:1, C18:0, C18:1, C18:2) and which also have a high economic value in food and feed industries [47], the industrial sector, pharmaceutical industries, and recycling [48,59]. Of the total BSF larvae fat composition, saturated fatty acids made up 49% (C12:0, C14:0, C16:0, C18:0) and unsaturated fatty acids made up 48% (C14:1, C16:1, C18:1, C18:2), of which 28% were MUFAs (C14:1, C16:1, C18:1) and 20% were PUFAs (C18:2).

We found that there was no previously published metabolic model of BSF which could utilize the genome sequence and the gene-protein-reaction (GPR) association. Hermetia_01 is the first metabolic model which is focused on specific fatty acid production through the implementation of diet uptake and biomass composition. The biomass objective function was used in the model to describe the organism’s necessary proliferation resources.

We performed different BSF resource allocation potential simulations to analyze (a) the specific growth and (b) fatty acid synthesis rates in larval fat bodies by doubling the glucose or EAA uptake rates in relation to our experimental dietary data reference.

As the BSF is a multi-tissue and multi-compartmented organism, the proposed metabolism simplification by model Hermetia_01 may not provide ideal optimization results; still, we believe that fatty acid production from carbohydrates in most organisms is synthesized by central metabolic biochemical pathways via acetyl-COA. We introduced detailed fatty acid production pathways by integrating lumped reactions [41]. These reactions represented all the metabolites involved in the fatty acid synthesis metabolic pathway which have manually corrected stoichiometry and are mass balanced.

Comparing the *FlySilico* [52] *D. melanogaster* with Hermetia_01 with regard to their efficacy as insect-based metabolic models, Hermetia_01 contains a more detailed description of TAG synthesis because it incorporates different saturated fatty acids (C12:0, C14:0, C16:0, C18:0) and unsaturated fatty acids (C14:1, C16:1, C18:1, C18:2) instead of a generalized TAG description, which lacks the specificity for lipid metabolism and resource allocation process modeling. Thus, Hermetia_01 is capable of predicting the impact of dietary changes on biomass production as well as the fatty acid resource allocation processes in the fat body during the larval development.

Given access to quality experimental omics data and using gene-protein-reaction association properties and steady-state assumptions, the model can be used to predict the necessary diet composition to achieve a desirable insect biomass composition and a specific growth rate. Model versions are available in the SBML, MAT, XLSX, and JSON formats, allowing the widespread use of various paid and open-access metabolic modeling tools and environments. We also provided the Hermetia_01 metabolic model biochemical pathway visual layout, which is compatible with an in-house-built web application for interactive metabolic flux analysis and the visualization tool IMFler [40]. Experts can easily perform the FBA or FVA metabolic analysis algorithms in IMFLer, utilizing the user-friendly interface on a local computer, with no need to install metabolic modeling tools.

The unbalanced fatty acid content of BSF larvae, however, is one of the main challenges faced in their industrial application [10]. Hoc et al. also indicate that fatty acids can be directly accumulated from the diet and via de novo synthesis in the BSF [14], suggesting the need for further experiments on amino acids and the role of the carbohydrate level in BSF diets. Our Hermetia_01 metabolic model already allows us not only to predict dietary carbon accumulation or resource allocation by de novo fatty acid biosynthesis but also to show the dietary EAA and carbohydrate impact on metabolism.

The Hermetia_01 model can predict how dietary glucose uptake changes affect specific fatty acid synthesis in the fat body during the larval development stage. We observed the positive impact of an increased glucose uptake rate on fatty acid synthesis (Figure 5), where the optimization results showed an increase of ~30% in fatty acid production rates due to a doubled glucose consumption. One exception was C18:2, which resulted in a 75.86% production increase as a result of the doubled glucose consumption. However, an optimal glucose uptake threshold should be determined in order to maintain high dietary carbon conversion efficiency. As the amount of glucose added to the diet during the in silico experiments was not used efficiently for fatty acid production (Figure 6), the surplus nutrition was diverted to other metabolic needs in the model.

Many experiments have been conducted in which the direct accumulation of fatty acids from the environment was observed but not the fatty acids synthesized by de novo biochemical pathways. The diet composition and rearing environmental conditions in most cases are not similar; thus, some reports have contradictory results, where the oleic and linoleic acid proportions can vary significantly [60,61]. Further investigation is necessary to better understand the fatty acid accumulation processes.

Our Hermetia_01 metabolic model optimization results also suggest that BSF significant growth rate changes can be achieved by increasing the dietary protein content, especially the EAA amounts (Figure 4), indicating that a glucose or other mono-sugar increase alone is not enough to improve the specific growth rate. Similar conclusions have been reported by other researchers studying the dietary impact on *H. illucens* larva metabolism and growth [57].

The novel Hermetia_01 metabolic model, as well as our *in silico* experiments (Appendix A), can be applied to the adjustment of the BSF larva diet composition to increase the biotechnological potential of different product metabolites and to determine the best waste products or residues for BSF larvae to recycle.

This is becoming especially fascinating as new human diets arise, such as the low-carbohydrate and keto diets. Insect farming requires fewer resources, has the potential to extract, accumulate, and reuse high-value nutrients from waste, and can be successfully employed in small-scale as well as large-scale production. Insect breeding requires less water for 1 kg compared to red meat and other conventional livestock practices. In this context, insect metabolic modeling is rapidly gaining interest and demand in scientific, food and feed, medical, and other biotechnological industries and applications.

## Figures and Tables

**Figure 1 metabolites-13-00724-f001:**
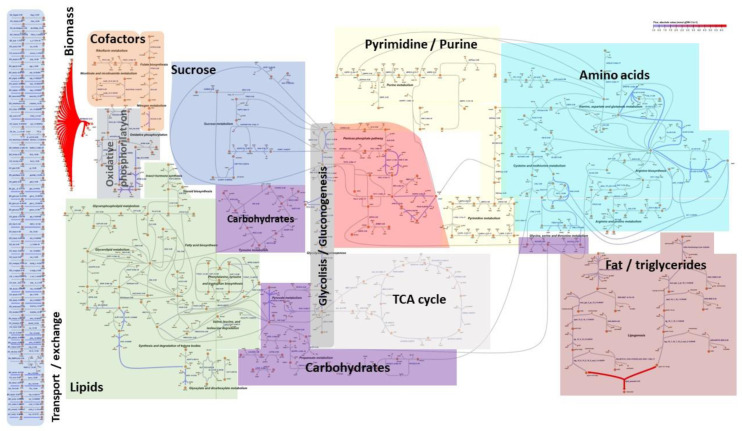
Metabolic model biochemical pathways of *H. illucens* for resource allocations analysis for high-value fatty acids accumulation. Each color represents a separate pathway. Biochemical network figure available in JPEG format (Appendix A).

**Figure 2 metabolites-13-00724-f002:**
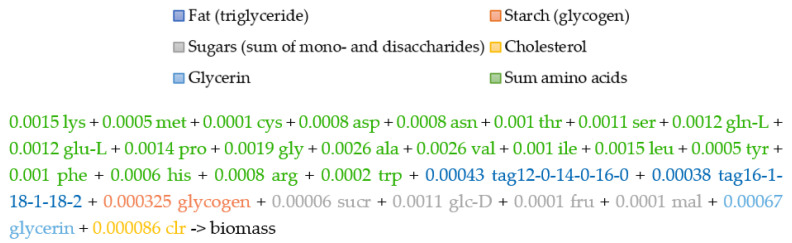
Biomass composition and function of BSF larva. Biomass objective function based on the chemical analysis experiments (Appendix A).

**Figure 3 metabolites-13-00724-f003:**
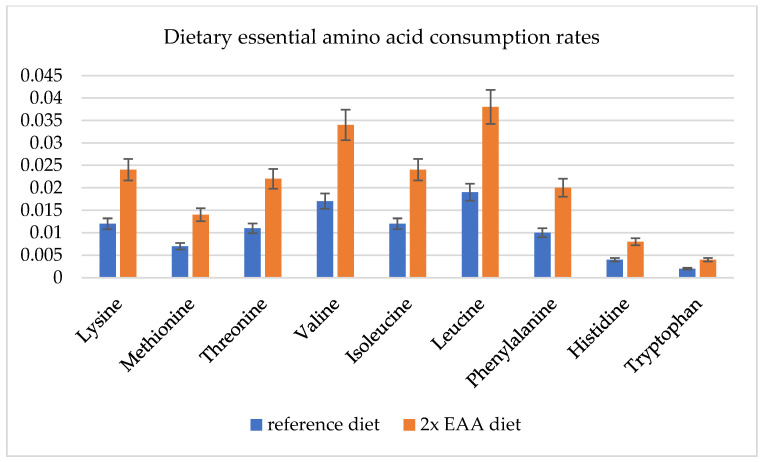
Dietary essential amino acid (EAA) consumption rates for reference diet and doubled EEA diet, mmol/larva/day. Optimization results in using only 90% of substrate amount in the model; thus, error bars are 10%.

**Figure 4 metabolites-13-00724-f004:**
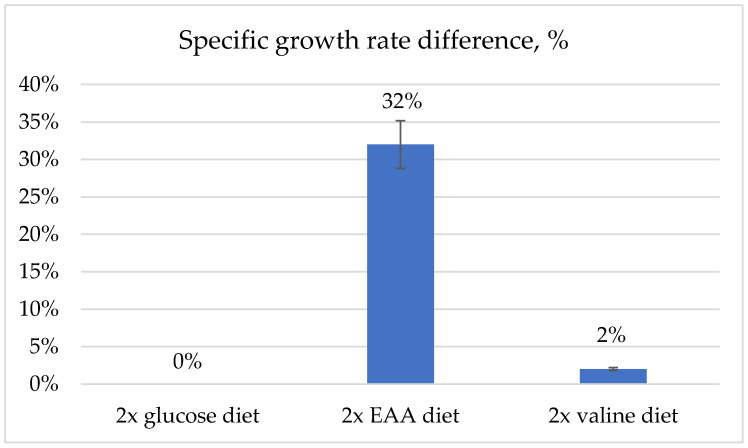
The predicted growth rate of BSF larvae with doubled glucose consumption diet; the doubled essential amino acid (EAA) consumption diet; and the doubled valine consumption diet. Comparisons are made based on our dietary experimental data (reference diet) and resulting growth rate prediction in the Hermetia_01 model; changes calculated in percent. Optimization results in using only 90% of substrate amount in the model; thus, error bars are 10%.

**Figure 5 metabolites-13-00724-f005:**
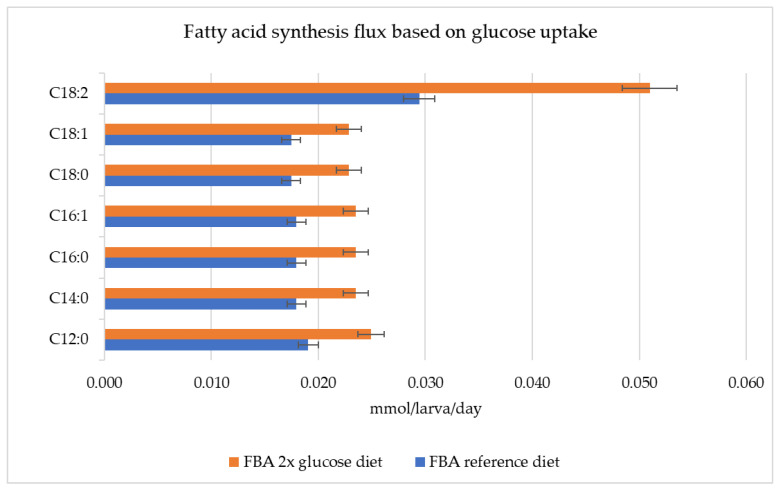
Fatty acid synthesis flux rates with reference diet and doubled glucose consumption diet, mmol/larva/day.

**Figure 6 metabolites-13-00724-f006:**
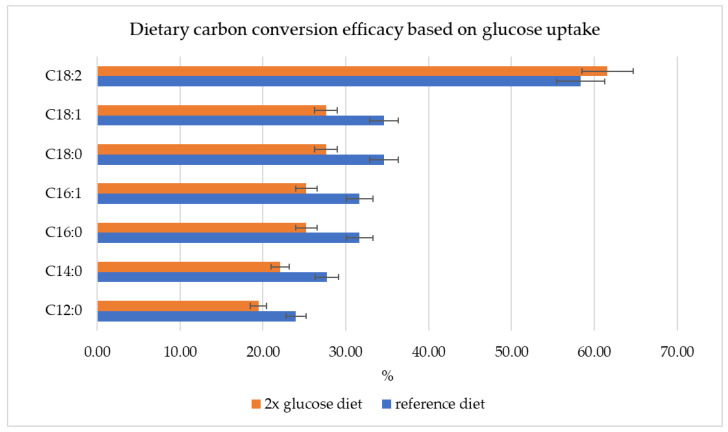
The dietary carbon conversion efficiency with reference diet and under the conditions of doubled glucose consumption.

**Table 1 metabolites-13-00724-t001:** Applied diet ingredients, ratios, and residue.

Ingredients	%	g/100 g	Total Diet, g	Left in Residue, g
Glucose	10	10.58	846	0
Egg White Protein	20	23.38	1870	1310
Water	70	66.05	5284	4890

**Table 2 metabolites-13-00724-t002:** Dietary amino acid profile conversion to transport reaction rates per 1 larva a day (detailed measurement data in Appendix A).

Amino Acid Profile	mg per 100 g Protein	Consumed Protein Weight, g	mg/larva/day	mmol/larva/day
Lysine	6000	85.272	1.769	0.012
Methionine	3500	50.116	1.039	0.007
Cystine	2500	36.465	0.756	0.003
Aspartate	4800	48.62	1.008	0.008
Asparagine	4800	48.62	1.008	0.008
Threonine	4300	62.832	1.303	0.011
Serine	6400	93.687	1.943	0.018
Glutamate	6150	62.271	1.292	0.009
Glutamine	6150	62.271	1.292	0.009
Proline	3600	52.921	1.098	0.010
Glycine	3300	47.872	0.993	0.013
Alanine	5800	84.898	1.761	0.020
Valine	6500	95.183	1.974	0.017
Isoleucine	5100	74.239	1.540	0.012
Leucine	8000	117.062	2.428	0.019
Tyrosine	3700	54.604	1.133	0.006
Phenylalanine	5500	78.914	1.637	0.010
Histidine	2200	32.164	0.667	0.004
Arginine	5400	79.475	1.648	0.009
Tryptophan	1500	23.001	0.477	0.002

**Table 3 metabolites-13-00724-t003:** Fatty acid chemical analysis results of *H. illucens* larvae.

Fatty Acid	g/100 g
Caprylic acid (C8:0)	0.13
* Lauric acid (C12:0)	29.4
* Myristic acid (C14:0)	5.75
Myristoleic acid (C14:1)	0.73
* Palmitic acid (C16:0)	10.45
* Palmitoleic acid (C16:1)	5.69
Stearic acid (C18:0)	3.5
* Oleic Acid (C18:1)	21.05
* Linoleic acid (C18:2)	19.8
Linolenic acid, alpha (C18:3)	0.48
Linolenic acid, gamma (C18:3)	<0.10
Eicosapentaenoic acid (C20:5)	<0.10
Docosahexaenoic acid (C22:6)	<0.10

* fatty acids which were included in Hermetia_01 metabolic model biomass composition function.

**Table 4 metabolites-13-00724-t004:** Metabolic network comparison.

Organism	Model ID	Metabolites	Reactions	Genes	% Dead-End Metabolites	% Blocked Reactions	% Unbalanced Reactions	% Exchange Reactions
*E. coli*	e_coli_core	72	95	137	4.21	17.51	16.84	21.10
*H. illucens*	Hermetia_01	326	407	471	2.45	16.71	19.66	18.43
*S. cerevisiae*	yeastGEM_v8.6.0	2749	4069	1151	12.44	31.73	11.40	6.46
*E. coli*	iWFL_1372	1973	2782	1372	9.17	40.08	12.80	14.06
*Human*	Recon3D_01	8399	13543	3697	6.56	11.70	17.19	13.97

The model studied in the article is marked in green.

## Data Availability

The current manuscript uses previously published *H. illucens* genome annotation data found in (https://www.ncbi.nlm.nih.gov/assembly/GCF_905115235.1 (accessed on 21 March 2023)). All experimental measurements and metabolic model and optimization results are included in the article. For additional information and follow-up studies please also visit (https://biomod.lv/ (accessed on 21 March 2023)).

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
