# Peer review of "Metabolic Modeling of Hermetia illucens Larvae Resource Allocation for High-Value Fatty Acid Production"

_metabolites, 2023, doi:10.3390/metabo13060724_

Round 1

Reviewer 1 Report

General comments: The authors performed metabolic modeling of black soldier fly larvae. The manuscript as written lacks clarity in both syntax and narrative. The model does not seem to be validated by experiments. The manuscript needs to be revised considerably before acceptable for publication. Additional comments, queries, and suggestions can be found in my specific comments.

 Specific comments

Lines 48-49: Italicize the scientific names.

Lines 83-94: These do not belong here.

Lines 96-101: These sentences do not belong to the method section.

Figure 1 is not very legible.

Line 256: “0.028 g”.

Line 150: Why did the authors measure starch content?

Line 204: “studied”.

Line 331: Starch and glycogen are different.

Figure 2: What are the numbers?

Table 1: Use “.” Instead of “,” for decimal point.

Line 344: These are not amino acids.

Figure 3: Where are the error bars? What do “ref_state” and “2x_EAA” mean?

Figure 4: Error bars missing.

Lines 419-422: Did the authors validate the in silico prediction?

Author Response

 Specific comments

Lines 48-49: Italicize the scientific names. (Italicized in lines 64-65)

Lines 83-94: These do not belong here. (Removed from Introduction section)

Lines 96-101: These sentences do not belong to the method section. (Lines relocated from Materials and methods section to Introduction lines 36-44)

Figure 1 is not very legible. (Available in Supplementary file 7)

Line 256: “0.028 g”. (Fixed in line 261)

Line 204: “studied”. Corrected

Line 150: Why did the authors measure starch content? We measured glycogen content. Corrected to “glycogen”.

Line 331: Starch and glycogen are different. (Corrected to “glycogen”)

Figure 2: What are the numbers? (Calculations for biomass precursor ratios can be found in Supplementary file 9, which is referenced in lines 323 and 332)

Table 1: Use “.” Instead of “,” for the decimal point. (Fixed in lines 351-352)

Line 344: These are not amino acids. (Changed to “fatty acids” in line 352)

Figure 3: Where are the error bars? What do “ref_state” and “2x_EAA” mean? (5% error bars added and legends clarified in lines 399-401)

Figure 4: Error bars missing. (5% error bars added in line 406)            

The authors did not do experimental field-rearing experiments based on metabolic model optimisation results, but the model’s data showed similar growth rate prediction correlates with latest experiments. Broeckx, L.; Frooninckx, L.; Slegers, L.; Berrens, S.; Noyens, I.; Goossens, S.; Verheyen, G.; Wuyts, A.; Van Miert, S. Growth of Black Soldier Fly Larvae Reared on Organic Side-Streams. Sustainability 2021, 13, 12953, doi:10.3390/su132312953.

van Huis, A.; Oonincx, D.G.A.B. The Environmental Sustainability of Insects as Food and Feed. A Review. Agron. Sustain. Dev. 2017, 37, 43, doi:10.1007/s13593-017-0452-8.

Reviewer 2 Report

This paper by Grausa et al. presents the first complete metabolic model for, Hermetia illucens, an insect widely used in industry for the recovery of various organic wastes. It is mainly focused on H. illucens production capabilities for some specific fatty acids of interests when the flies larvae are fed with two different diets.

The paper is globally well written and well documented allowing other searchers to handle the model quite easily. I have made few suggestions (see below) that to my opinion could ameliorate the reading of the paper.

The authors tested 2 diets as compared to a reference: 2x glucose et 2x amino acids. I think that they could have calculated a C/N ratio of the different diets that should be discussed. Moreover, a more detailed analysis should be included concerning amino acid assimilation (degradation and interconversion) as it is crucial in this context.

Below are my comments all along the text:

L50-59: mostly auto-citation not really in the topic of the paper. That part may be shortened.

L80-94: I don’t like very much when results are detailed in the introduction, here this part can be shortened.

L96-110: to my opinion, this part should be placed in introduction. Where the questions the diets including C/N ratio should be evoked.

L183: E. coli (italicize)

L184: supplementary (“i” should be corrected) – sup 6 is cited before the other sup. Moreover there are very confusing numbering (N°7 name of the file, and is written N°5 inside the file…) and correct “Glycolysis” inside the file.

L203: usually nucleic acid biosynthesis is included in the biomass objective function. Why it is not the case here? At least the author should explain why nucleic acid biosynthesis was neglected (or indirectly integrated) in the model.

L283: Ci

L290: in the sup1 a sup 10 is described (pb of numbering). Correct “chemical and experimental” (remove “s”)

L335 and 343: table 3 (and decenter the lower note of the table and replace “amino” by “fatty”

L347-353: might be placed in the introduction.

L360-365: These results are of importance and the authors say that they are discussed in the Metabolic network comparison, but I couldn’t find this section. To my opinion this is a compulsory point to correct.

L389: Figure 3 is not the right figure to cite here.

L393: I don’t see the need of this figure (I don't understand what it is). I don’t believe that if you feed larvae with twice concentration of amino acids, the global composition of amino acid inside larvae will be double (otherwise you are only following the stomachal content)? More explanations are needed.

L398: Why the model cannot simulate all the amino acids degradations (the scheme is not so complexe)? If it is the case, this is a big lack of the model especially to address this question of C/N ratio in the diet. To my opinion this analysis should be developed, and all amino acid consumption should be studied.

L414-415: the “double” is not see on figure 6 (but on figure 5), the expression and the deduction is a bit confusing.

L525: sup 8 to 10?

L546: All the species names should be italicized.

Author Response

The authors tested 2 diets as compared to a reference: 2x glucose et 2x amino acids. I think that they could have calculated a C/N ratio of the different diets that should be discussed. Moreover, a more detailed analysis concerning amino acid assimilation (degradation and interconversion) should be included as it is crucial in this context.

The goal of our research was to build the first metabolic model of Hermetia illucens for resource allocation in the larva, where we assumed that all carbon is accumulated in the biomass and fatty acids. Experimental diet and larva composition measurements were focused on carbon bioconversion; thus, nitrogen assimilation or bioconversion was not raised as the research aim. For analyzing carbon allocation in H. illucens larvae as an example we used the Hatzimanicatis published systematic reduction and analysis of genome-scale metabolic models’ method (https://doi.org/10.1371/journal.pcbi.1005444) to build the first H. illucens metabolic model in a simplified form. Below are my comments all along the text.

L50-59: mostly auto-citation not really in the topic of the paper. That part may be shortened.

Done ! (L66)

L80-94: I don’t like very much when results are detailed in the introduction, here this part can be shortened. (Removed from Introduction section)

L96-110: to my opinion, this part should be placed in introduction. (Lines relocated from Materials and Methods section to Introduction lines 36-44) Where the questions the diets including C/N ratio should be evoked.

L183: E. coli (italicize) (Italicized in line 185)

L184: supplementary (“i” should be corrected) (Corrected in line 187) – sup 6 is cited before the other sup. Moreover there are very confusing numbering (N°7 name of the file, and is written N°5 inside the file…) (Supplementary file 7 title edited) and correct “Glycolysis” inside the file.

L203: usually nucleic acid biosynthesis is included in the biomass objective function. Why it is not the case here? At least the author should explain why nucleic acid biosynthesis was neglected (or indirectly integrated) in the model.

Fat and protein were the main contributors to the H. illucens dry biomass. Since the authors couldn’t measure nucleic acids’ macromolecular component for biomass objective function due to a lack of nucleic acids measurement equipment, thus authors decided that the use of different organisms’ nucleic acids macromolecular values in biomass objective function will generate faulty optimisation results.

L283: Ci (Corrected in line 289)

L290: in the sup1 a sup 10 is described (pb of numbering). Correct “chemical and experimental” (remove “s”) (Corrected in supplementary file 1)

L335 and 343: table 3 (and decenter the lower note of the table and replace “amino” by “fatty” (Corrected in line 352)

L347-353: might be placed in the introduction. (Relocated from Results section 3.3 to Introduction lines 99-106)

L360-365: These results are of importance and the authors say that they are discussed in the Metabolic network comparison, but I couldn’t find this section. To my opinion this is a compulsory point to correct. (Comparison table added in line 372)

L389: Figure 3 is not the right figure to cite here. (Reference changed to table 4 in line 393)

L393: I don’t see the need of this figure (I don't understand what it is). I don’t believe that if you feed larvae with twice concentration of amino acids, the global composition of amino acid inside larvae will be double (otherwise you are only following the stomachal content)? More explanations are needed. (Wrong figure titles were added, figure 3 shows dietary EAA consumption rates, is corrected in lines 399-401)

L398: Why the model cannot simulate all the amino acids degradations (the scheme is not so complexe)? If it is the case, this is a big lack of the model especially to address this question of C/N ratio in the diet. To my opinion this analysis should be developed, and all amino acid consumption should be studied.

Due to the complexity of H illucens metabolism and the lack of equipment to measure all larva biomass macromolecular composition the research aim was to study the resource allocation of carbon in the organism. As the model is the first version of a multi-tissue and multi-compartmented organism, we do not include nitrogen, phosphate, oxygen, carbon dioxide and other chemical elements impact analysis on resource allocation processes. We focused more on fatty acids and biomass growth rate analysis. The next version of the H illucens metabolic model will be based on a genome-scale metabolism and will include much more secondary metabolites production capacities and thus could be used to analyse C/N ratio impact on h. illucens larva in different developmental stages.

L414-415: the “double” is not see on figure 6 (but on figure 5), the expression and the deduction is a bit confusing. (Description corrected in lines 418-424)

L525: sup 8 to 10?

Supplementary materials 8 – line 249

Supplementary materials 9 – line 310

Supplementary materials 10 – line 504

Reviewer 3 Report

Metabolic modeling of Hermetia illucens larvae resource allocation for high-value fatty acid production

The paper is very innovative. These models can help to standardize the black soldier fly breeding in terms of nutritional value depending on substrates. Anyway, the discussion should be reduced in some parts. Some minor revisions were added to the paper

Introduction:

Line 40-41: “BSF has gained significant interest in food and feed, biofuel, pharmaceutical, lubricants and fertilizer sciences and industrial applications.”

Please, add more specific references; I suggest for lipids application a review:

-        https://doi.org/10.3390/su131810198

Line 75-76: “Saturated fatty acid (C12:0) the content in BSF larvae biomass was 29% of the total fatty acid profile”

Please, specify which is the fatty acid in question and don’t write saturated fatty acid. I suggest “lauric acid (C12:0)” and for the next text you can use the abbreviation. This should be for every fatty acid.

Materials and methods:

Please, this section should contain only information only on the experiments. Others should be written in introduction or discussion.

Line 182: “…the model scale includes extended central metabolism pathways analogue to the functionality of E. coli…”

Please, write the name of bacteria and insects in italic. This is for every name of bacteria and insects

RESULTS

Line 246: Table 4 is not reported in the paper

Line 247: Table 3 is not reported in the paper

Author Response

The paper is very innovative. These models can help to standardize the black soldier fly breeding in terms of nutritional value depending on substrates. Anyway, the discussion should be reduced in some parts. Some minor revisions were added to the paper

Introduction:

Line 40-41: “BSF has gained significant interest in food and feed, biofuel, pharmaceutical, lubricants and fertilizer sciences and industrial applications.”

Please, add more specific references; I suggest for lipids application a review:

-        https://doi.org/10.3390/su131810198

Corrected in line 51

Line 75-76: “Saturated fatty acid (C12:0) the content in BSF larvae biomass was 29% of the total fatty acid profile”

Please, specify which is the fatty acid in question and don’t write saturated fatty acid. I suggest “lauric acid (C12:0)” and for the next text you can use the abbreviation. This should be for every fatty acid. (Corrected in lines 91-95)

Materials and methods:

Please, this section should contain only information only on the experiments. Others should be written in introduction or discussion. (First two paragraphs relocated to Introduction section)

Line 182: “…the model scale includes extended central metabolism pathways analogue to the functionality of E. coli…”

Please, write the name of bacteria and insects in italic. This is for every name of bacteria and insects (Corrected in line 185)

RESULTS

Line 246: Table 4 is not reported in the paper (Corrected line 249)

Line 247: Table 3 is not reported in the paper (Corrected line 249)

Reviewer 4 Report

The manuscript titled “Metabolic modeling of Hermetia illucens larvae resource allocation for high-value fatty acid production” reports on a new metabolic modeling design approach to predict how different nutrients in the diet of the Black Soldier Fly larvae influence their larval proximate composition with particular emphasis on the fatty acid profile. As such the ms is of great interesting to a broad spectrum of readers as it offers new insights into the comprehension of resource allocation in the larval stage of one of the most promising insect species for a wide range of applications.

Overall, the ms is well written and organized and presents robust results and discussions, often supported by relevant references.

The research covers a current topic, especially for the great potential of BSFL in the waste valorisation process though the bioconversion of different organic waste to produce valuable larval biomass and useful by-products in the context of a circular economy.

Just a few minor points need to be fixed before publication.

In particular, I would invite the authors to fix the citations in the text according to the guidelines for authors “In the text, reference numbers should be placed in square brackets [ ], and placed before the punctuation; for example [1], [1–3] or [1,3]”; check line 34, 43, 51, 54, 99, 104, etc.

The name of the species must be italicized; fix Escherichia coli and Saccharomyces cerevisiae (lines 48-49, 143).

In the Introduction section, the goals of the study are clear, but I am not sure the obtained results should be reported in this section; please check the guidelines of the journal.

In the Materials and methods section, some paragraphs report the verbs in the present tense (please fix sub-sections 2.3.2, 2.3.4, 2.3.5). Two aspects need further clarifications: 1. how did the authors calculate the dry weight of the larvae? The results of fatty acids and EAA are reported in a dry matter basis? 2. Did the larval diets influence the growth of the larvae at these early developmental stages? A different diet can determine a significant shift among the six larval developmental stages. After the specific 9-day diet, BSF larvae were at the V instar larvae or prepuapae stage?

The outcomes give new tools to predict the impact of dietary changes on biomass production that is especially interesting in the perspective of using BSFL as feed ingredient, food and in other industrial applications in the next future.

Minor points

The name of the species is not always italicized. Please check throughout the text and reference list

Line 76: delate “the” in “(C12:0) the content”

Line 143: in n-hexane “n” should be italicized

Lines 452-453 are not clear

Lines 470-471: please choose either larval development or developmental stages

Author Response

Just a few minor points need to be fixed before publication.

In particular, I would invite the authors to fix the citations in the text according to the guidelines for authors “In the text, reference numbers should be placed in square brackets [ ], and placed before the punctuation; for example [1], [1–3] or [1,3]”; check line 34, 43, 51, 54, 99, 104, etc. (Corrected)

The name of the species must be italicized; fix Escherichia coli and Saccharomyces cerevisiae (lines 64-65, 185). (Corrected)

In the Introduction section, the goals of the study are clear, but I am not sure the obtained results should be reported in this section; please check the guidelines of the journal. (Removed from Introduction section)

In the Materials and methods section, some paragraphs report the verbs in the present tense (please fix sub-sections 2.3.2, 2.3.4, 2.3.5). Two aspects need further clarifications:

The results of fatty acids and EAA are reported in a dry matter basis?

the fresh weight of the larvae calculation included (line  258 - 261)

The results of fatty acids and EAA are reported in a dry matter basis

  1. Did the larval diets influence the growth of the larvae at these early developmental stages?

Diet was not changed during the rearing process. The authors started obtaining experimental measurements on the 9th day and continued rearing for 7 more days, thus we didn’t obtain early developmental stage data, however, our metabolic model predicts changes in growth when the diet is altered.

 A different diet can determine a significant shift among the six larval developmental stages. After the specific 9-day diet, BSF larvae were at the V instar larvae or prepuapae stage?

BSF larvae take approximately two weeks to reach the prepupal stage. So, after 9-day specific diet, they were at V instar larvae stage. (Lines 109-112)

The outcomes give new tools to predict the impact of dietary changes on biomass production that is especially interesting in the perspective of using BSFL as feed ingredient, food and in other industrial applications in the next future.

Minor points

The name of the species is not always italicized. Please check throughout the text and reference list (Corrected)

Line 76: delate “the” in “(C12:0) the content” (Corrected in line 91)

Line 143: in n-hexane “n” should be italicized (Corrected in line 142)

Lines 452-453 are not clear (Corrected in lines 463-464)

Lines 470-471: please choose either larval development or developmental stages (Corrected in line 482)

Round 2

Reviewer 1 Report

The authors have addressed the reviewers’ comments and revised the manuscript accordingly. I still have a question for the error bars in Figures 3 & 4. What did the authors mean by “5% error bars added”? Why not use standard deviation?

Author Response

All the results of computational simulations results show the most optimal solution for a product (fatty acids and specific growth) maximal flux in mmol/larva/day. The computational simulations were performed using the Cobra Toolbox flux variability analysis function considering only the solutions, which give at least 90% of the product (fatty acids, growth rate) maximal value. 10 % of flux distribution is considered an error bar for computational simulation results.

The authors corrected the error bar value from 5% to 10%.